# Hidden noise in immunologic parameters might explain rapid progression in early-onset periodontitis

George Papantonopoulos[1]*, Chryssa Delatola[2], Keiso Takahashi[3], Marja L. Laine[2], Bruno G. Loos[2]

1 Center for Research and Applications of Nonlinear Systems, Department of Mathematics, University of Patras, Patras, Greece, 2 Department of Periodontology, Academic Center for Dentistry Amsterdam (ACTA), University of Amsterdam and Vrije Universiteit Amsterdam, Amsterdam, The Netherlands, 3 Department of Conservative Dentistry, School of Dentistry, Ohu University, Fukushima, Fukushima, Japan

* ppntnpls@otenet.gr

**Editor:** Özlem Yilmaz, Medical University of South Carolina, UNITED STATES

**Data Availability Statement:** All relevant data are within the manuscript and its Supporting Information files.

## Abstract

To investigate in datasets of immunologic parameters from early-onset and late-onset periodontitis patients (EOP and LOP), the existence of hidden random fluctuations (anomalies or noise), which may be the source for increased frequencies and longer periods of exacerbation, resulting in rapid progression in EOP. Principal component analysis (PCA) was applied on a dataset of 28 immunologic parameters and serum IgG titers against periodontal pathogens derived from 68 EOP and 43 LOP patients. After excluding the PCA parameters that explain the majority of variance in the datasets, i.e. the overall aberrant immune function, the remaining parameters of the residual subspace were analyzed by computing their sample entropy to detect possible anomalies. The performance of entropy anomaly detection was tested by using unsupervised clustering based on a log-likelihood distance yielding parameters with anomalies. An aggregate local outlier factor score (LOF) was used for a supervised classification of EOP and LOP. Entropy values on data for neutrophil chemotaxis, CD4, CD8, CD20 counts and serum IgG titer against *Aggregatibacter actinomycetemcomitans* indicated the existence of possible anomalies. Unsupervised clustering confirmed that the above parameters are possible sources of anomalies. LOF presented 94% sensitivity and 83% specificity in identifying EOP (87% sensitivity and 83% specificity in 10-fold cross-validation). Any generalization of the result should be performed with caution due to a relatively high false positive rate (17%). Random fluctuations in immunologic parameters from a sample of EOP and LOP patients were detected, suggesting that their existence may cause more frequently periods of disease activity, where the aberrant immune response in EOP patients result in the phenotype "rapid progression".

## Introduction

Periodontitis is a complex disease with multiple causal factors (bacteria and viruses, life style, (epi)genetic background, systemic diseases, tooth and dentition related and most likely

**Funding:** The authors received no specific funding for this work.

**Competing interests:** The authors have declared that no competing interests exist.

stochastic factors) interacting simultaneously in an unpredictable and nonlinear manner [1–3]. However, as local interactions are in general chaotic (sensitive to initial conditions and aperiodic), the system i.e. the disease, eventually evolves and self-organizes; it results in the ultimate emergence of a pattern that allows us to evaluate the system using statistical methods and mathematical modelling [4–6].

The old classification scheme for two decades recognized two clinical forms of periodontitis: chronic (CP) and aggressive (AgP) periodontitis [7]. The identification of AgP cases was based on rapid attachment loss and bone destruction, the absence of systemic factors to explain this progression rate and familial aggregation [8]. The age of 35 years was used arbitrarily as a cut-off point to discriminate between AgP and CP [9]. However, AgP and CP share genetic and other risk factors and it has been long recognized that cases of AgP can occur also in people aged over 35 years and that cases of CP can occur in people below this age [8–10]. An aberrant immune response (hypo- or hyper-response and/or lack of resolution) has been described to associate with advanced periodontitis, irrespective of being AgP or CP [1,2]. Also, limited differences between the gingival tissue transcriptional profiles of AgP and CP have been reported [11]. There is little consistent evidence that AgP and CP are different diseases [12]. The new periodontitis classification scheme [13] recognizes AgP and CP as one entity with 4 stages of severity and 3 grades of prognosis. Empirical evidence-driven thresholds of attachment loss were used to differentiate levels of periodontitis severity [14], while grades recognize risk factors that influence periodontitis progression and classify initially patients by a history-based analysis as patients with slow (grade A), moderate (grade B) and rapid progression rate (grade C).

The immune response to the invading periodontal pathobionts and viruses triggers a nonlinear destructive process for periodontal ligament and alveolar bone loss [2,15]. Nonlinearity means that a small change in them may have disproportionally large effects on their final behavior. Random fluctuations in a complex system are found inevitable. Their significance to gene expression and cell function are well recognized [16], however, they have not yet been explored in the pathogenesis of periodontitis.

In biological systems random fluctuations (also called anomalies or noise) might be responsible for certain phenotypes, as added anomalies to a nonlinear system might change its behavior with unexpected aberrant activity [17,18]. It is often observed in bistable systems, i.e. the existence of two stable states, such as the alternation between periods of exacerbation and remission in susceptible and chronically diseased subjects [19]. There is evidence that a small part of the population exhibits severe periodontitis while the majority of patients show mild to moderate periodontitis [20]. In a longitudinal study on a sample of unlabeled periodontitis patients followed over 5–8 years [6], we found possible evidence of two groups of patients on the basis of longitudinal radiographic bone loss. One out of 5 patients showed almost 5 times higher progression rate. Gene networks can generate bistable states [17] and bistability is a finding that supports the importance of random fluctuations (noise) to the emergence of a phenotype of periodontitis with rapid progression rate.

We hypothesize that random fluctuations in immunologic parameters of periodontitis patients might constitute the host response extra vulnerable to the bacterial challenges and might explain more frequent and longer periods of exacerbation resulting in the advanced tissue destruction found in the rapid progressive form with severe breakdown (new classification stages 3 or 4, grade C [13]) i.e. often the early-onset form of periodontitis (EOP). We aimed to investigate this hypothesis on a group of EOP and late-onset periodontitis patients (LOP), who–based on disease history–are characterized as either having a rapid progression rate (EOP stage 3–4, grade C) or having a slow progression rate (LOP stage 3, grade A). Another group of severe periodontitis patients suspected for EOP (i.e. grade C) served as a validation cohort.

# Results

Patient demographic (Table 1) and other characteristics have been described before [5–6]. The validation cohort has also been described and presented in a previous publication [21]. Table 2 presents the data for immunologic parameters. Mean values of IL-1, IL-4, IFN-γ and IgG titer for *C.o.* were statistically significantly lower in LOP compared to EOP, whereas CD8, CD20, CD4/CD8 ratio and IL-2 and were significantly higher in LOP compared to EOP. The remainder of the immunologic parameters did not show differences between EOP and LOP patients (Table 2).

The workflow for the final detection of a "rapid progression" phenotype is presented in Fig 1. Principal component analysis (PCA) showed IgG titer against *P.g.* (SU63), monocyte IL-2 production, CD3 lymphocyte counts, IgG titer against *P.g.* (FDC381) and monocyte IL-4 production as the principal components explaining 75% of the variance in the aggregate EOP and LOP sample. The subspace analysis aimed at identifying anomalies in the parameters that contribute zero at explaining the variance of the dataset (showing eigenvalue 0 in the scree plot of the PCA analysis) (Fig 2). There were 17 parameters comprising the residual PCA-subspace. They were leukocyte adhesion and neutrophil chemotaxis test results, CD4, CD8, CD20 lymphocyte counts and CD4/CD8 ratio, IFN-γ and IL-1 monocyte production and IgG titers against *E.c.*, *P.i.*, *P.n.*, *F.n.*, *T.d.*, *C.o.*, *A.a.* (Y4), *A.a.* (ATCC29523) and *A.a.* (SUNY67). These 17 parameters were evaluated for anomalies in their structure, firstly by sample entropy estimation and secondly by clustering importance by the two-step clustering method.

Entropy values indicated possible data anomalies for neutrophil chemotaxis, CD4, CD8 and CD20 counts and IgG titer against *A.a.* (ATCC29523) that might explain more regularly occurring disease exacerbations in EOP patients than in LOP patients (Table 3). These 5 parameters showed squared entropy values ≥3 (Table 3). Based on the second step of the unsupervised clustering of patients into two groups, we found for these five latter parameters a low clustering importance, also indicating that these parameters are possible sources of anomalies (Table 3, Fig 3). Sample entropy values in the validation cohort showed for these five parameters squared entropy values from 0.15 to 0.76, except for neutrophil chemotaxis that showed a squared entropy value 1.9, being the highest in this cohort with the possible highest value at 2.92 (Table 3). Thus the latter results indicate neutrophil chemotaxis as a parameter with possible anomalies in the validation cohort.

The distribution of local outlier (LOF) scores is given in Fig 4. We found 32% of LOP patients to score between 2 and 2.7, while 35% of EOP patients scored between 3.5 and 4.1 (Fig 4A). By separating localized from generalized EOP patients we found LOF score distributions to be similar in the two categories, with the generalized EOP category showing a higher maximum value (Fig 4B). Using the identified 5 predictor parameters in the subspace, i.e.

**Table 1. Demographics of the study population.**

|  | Total number | Gender male/female | Age Mean years ± SD |
|---|---|---|---|
| EOP[a]-localized | 18 | 6/12 | 19.9 ± 6.5 |
| EOP-generalized | 50 | 13/37 | 28.3 ± 5.8 |
| LOP[b] | 43 | 17/26 | 47.0 ± 11.0 |
| Validation cohort (EOP-suspected) | 51 | 12/39 | 36.0 ± 9.2 |

a. Early-onset periodontitis

b. Late-onset periodontitis

**Table 2. Median values [means ± standard deviations] of immunologic parameters and IgG [a] titers for patients with late-onset periodontitis (LOP) or early-onset periodontitis (EOP), as well as in patients of the validation cohort.** Comparisons between LOP and EOP were made by the Mann-Witney U test (in bold statistically significant results). Data derived from a previous study [21].

| Parameter | Late-onset periodontititis (N = 43) | Early-onset periodontitis (N = 68) | Validation Cohort (N = 51) |
|---|---|---|---|
| Neutrophil function | | | |
| Chemotaxis [b] | 52.60 [56.64 ± 28.74] | 42.15 [44.71 ± 17.84] | 42.00 [40.17 ± 15.65] |
| Phagocytosis [c] | 4.27 [4.83 ± 3.25] | 2.84 [6.89 ± 17.95] | 4.33 [4.91 ± 2.32] |
| Adhesion [d] | 71.19 [71.76 ± 7.77] | 60.41 [60.22 ± 19.24] | 70.20 [69.2 ± 8.33] |
| Lymphocyte subsets | | | |
| CD3 (%) [e] | 74.00 [61.03 ± 9.70] | 65.70 [65.09 ± 12.05] | 65.20 [62.31 ± 13.35] |
| CD4 (%) | 39.00 [41.63 ± 7.51] | 36.60 [37.28 ± 10.98] | 39.40 [38.16 ± 9.26] |
| **CD8 (%) [f]** | **28.60 [29.07 ± 6.62]** | **25.80 [25.02 ± 6.09]** | 21.60 [24.06 ± 6.49] |
| **CD20 (%) [g]** | **12.30 [16.38 ± 9.33]** | **9.95 [13.13 ± 4.41]** | 10.90 [11.08 ± 7.82] |
| **CD4/CD8 ratio [g]** | **2.40 [2.38 ± 0.86]** | **1,42 [1.62 ± 0.76]** | 1.60 [1.70 ± 0.62] |
| Cytokine productivity | | | |
| **IL-1 (pg/ml) [h]** | **3.50 [5.32 ± 3.83]** | **99.00 [436.72 ± 897.76]** | 114.5 [422.38 ± 813.33] |
| **IL-2 (pg/ml)** | **80.00 [118.40 ± 104.59]** | **3,40 [3.56 ± 1.94]** | 3.8 [8.42 ± 19.21] |
| **IL-4 (pg/ml)** | **3.90 [4.30 ± 3.88]** | **7.80 [9.01 ± 6.70]** | 7.70 [7.93 ± 2.25] |
| IL-6 (pg/ml) | 473.00 [503.20 ± 616.80] | 100.00 .[1957.74 ± 4944.28] | 242.00 .[2089.50 ± 4083.09] |
| TNF-α (pg/ml) [i] | 16.65 [42.93 ± 54.48] | 274.70 [358.20 ± 383.60] | 437.50 [712.83 ± 628.82] |
| **IFN-γ (pg/ml) [j]** | **9.70 [11.41 ± 6.01]** | **32.30 [109.27 ± 232.45]** | 12.35 [11.87 ± 5.52] |
| T-cell blastogenesis | | | |
| Anti-CD3 (dpm x $10^{-4}$) | 13.90 [15.96 ± 3.14] | 8.90 [12.96 ± 11.63] | 13.50 [13.54 ± 5.79] |
| PWM (dpm x $10^{-4}$) [k] | 6.50 [7.48 ± 4.59] | 5.60 [8.39 ± 7.65] | 8.60 [9.77 ± 6.36] |
| Serum IgG titers (ELISA units) | | | |
| A.a. (Y4) [l] | 0.57 [.67 ± 2.51] | 0.33 [0.43 ± 1.12] | -0.60 [1.05 ± 3.43] |
| A.a. (ATCC29523) | 0.40 [0.84 ± 1.08] | 0.21 [1.36 ± 2.88] | 0.07 [4.84 ± 23.84] |
| A.a. (SUNY67) | 0.68 [0.51 ± 0.49] | 0.54 [1.42 ± 2.59] | .-0.18 [0.21 ± 0.84] |
| **C.o. (S3) [m]** | **0.24 [0.01 ± 0.40]** | **-0.09 [0.11 ± 0.45]** | 1.00 [0.76 ± 5.96] |
| E.c. (ATCC23834) [n] | 0.08 [0.22 ± 0.48] | 0.45 [1.04 ± 1.95] | -0.11 [0.11 ± 0.41] |
| F.n. (ATCC25586) [o] | -.06 [0.68 ± 4.74] | 0.33 [3.70 ± 9.71] | -0.04 [1.06 ± 4.07] |

(Continued)

**Table 2.** (Continued)

| Parameter | Late-onset periodontititis (N = 43) | Early-onset periodontitis (N = 68) | Validation Cohort (N = 51) |
|---|---|---|---|
| *P.i.* (ATCC25611) [p] | -0.17 [-0.27 ± 0.18] | -0,15 [0.41 ± 1.61] | -0.13 [-0.01 ± 0.46] |
| *P.n.* (ATCC33563) [q] | 0.60 -[0.26 ± 1.16] | 0.15 [0.53 ± 1.56] | 0.45 [0.30 ± 0.95] |
| *P.g.* (FDC381) [r] | 1.59 [4.19 ± 4.90] | 2.98 [7.84 ± 218.07] | 1.54 [6.31 ± 12.73] |
| *P.g.* (SU63) | 0.52 [2.23 ± 4.35] | 1.41 [6.41 ± 19.58] | 1.01 [2.18 ± 4.65] |
| *T.d.* (ATCC35405) [s] | -.05 [0.12 ± 0.39] | 0.23 [0.93 ± 1.85] | 1.27 [0.63 ± 1.60] |
| *W.s.* (ATCC29543) [t] | 0.37 [0.88 ± 0.99] | 0.33 [14.72 ± 56.08] | 0.35 [5.60 ± 14.31] |

[a] Ig = immunoglobulin

[b] Number of neutrophils migrated

[c] Number of bacteria internalized by 100 neutrophils

[d] Number of neutrophils adhered

[e] CD = cluster of differentiation

[f] Significantly different between EOP and LOP, p = 0.008

[g] Significantly different between EOP and LOP, p = 0.007

[h] IL = interleukin, significantly different between EOP and LOP for IL-1 and IL-2, p = 0.0001

[i] TNF-α = tumor necrosis factor

[j] IFN-γ = interferon, significantly different between EOP and LOP, p = 0.0001

[k] PWM = pokeweed mitogen

[l] *A.a.* = *Aggregatibacter actinomycetemcomitans*

[m] *C.o.* = *Capnocytophaga ochracea*, significantly different between EOP and LOP, p = 0.018

n *E.c.* = *Eikenella corrodens*

[o] *F.n.* = *Fusobacterium nucleatum*

[p] *P.i.* = *Prevotella intermedia*

[q] *P.n.* = *Prevotella nigrescens*

[r] *P.g.* = *Porphyromonas gingivalis*

[s] *T.d.* = *Treponema denticola*

[t] *W.s.* = *Wolinella succinogens*

neutrophil chemotaxis, CD4, CD8, CD20 counts and IgG titer against *A.a.* (ATCC29523), for an aggregated LOF, gave 94% sensitivity and 83% specificity in identifying EOP by a k-NN classifier (k = 5 chosen by 10-fold cross-validation), but with lower sensitivity in a 10-fold cross-validation (CV) of the model (87% sensitivity and 83% specificity).

## Discussion

We aimed to detect anomalies (random fluctuations) in immunologic parameters from a sample of EOP (stage 3–4, grade C) and LOP patients (stage 3, grade A). We aggregated the two samples to perform LOF measurements that could possibly discriminate EOP from LOP. PCA found IgG titer against *P.g.* (SU63), monocyte IL-2 production, CD3 lymphocyte counts, IgG titer against *P.g.* (FDC381) and monocyte IL-4 production as principal components in explaining the variance of the aggregate EOP and LOP sample. On the opposite side, the analysis on the PCA-subspace parameters suggested evidence for anomalies in neutrophil chemotaxis, CD4, CD8, CD20 counts and serum IgG titers against *A.a.*, that might explain more regularly

## Workflow of the procedures involved in the study

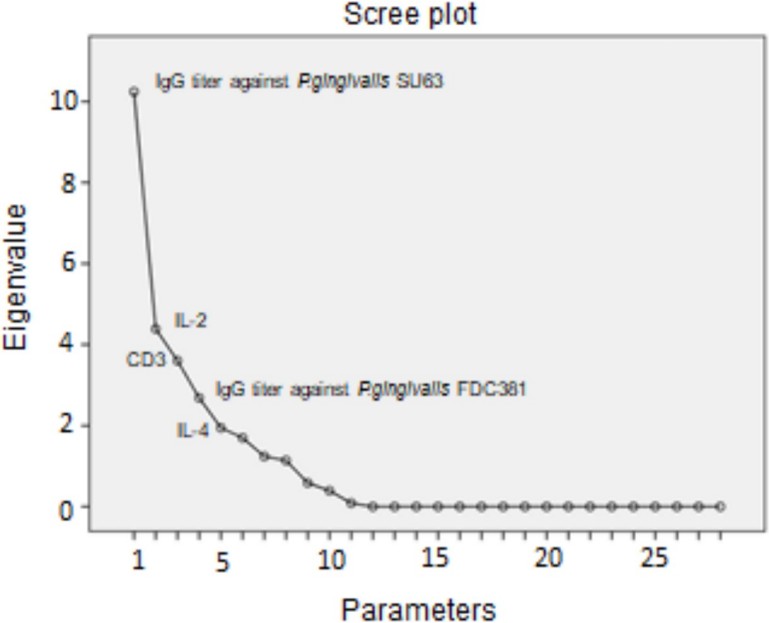

**Fig 1. Workflow to detect the "rapid progression" phenotype.** Immunologic parameters of early-onset periodontitis with rapid progression (EOP) and late-onset periodontitis (LOP) patients are aggregated for a principal component analysis (PCA) to identify the sub-space parameters and subsequently to calculate sample entropy and clustering importance for these parameters. We end up with a supervised classification of EOP and LOP patients.

**Fig 2. Finding normal and residual principal component analysis subspaces.** The eleven first principal components delineate the normal subspace, where almost 100% of the total variance is explained. The rest 17 parameters at eigenvalue 0 comprise the residual subspace where possible hidden anomalies might be found. They were leukocyte adhesion and neutrophil chemotaxis test results, CD4, CD8, CD20 and CD4/CD8 lymphocyte counts, IFN-γ and IL-1 monocyte production and IgG titers against *E.c.*, *P.i.*, *P.n.*, *F.n.*, *T.d.*, *C.o.*, *A.a.* (Y4), *A.a.* (ATCC29523) and *A.a.* (SUNY67).

**Table 3. Anomaly detection in the 17 parameters of the residual Principal Component Analysis (PCA) subspace by high sample entropy or low unsupervised clustering importance scores.** Detected parameters with possible anomalies are in bold.

| | Discovery cohort | | Validation cohort |
|---|---|---|---|
| | Squared entropy Possible range (0–4.18) | Clustering importance score | Squared entropy Possible range (0–2.92) |
| Cellular immune parameters | | | |
| Leukocyte adhesion | 2.44 | 0.09 | 1.60 |
| **Neutrophil chemotaxis** | **3.91** | **0.05** | **1.90** |
| **CD4 lymphocytes** [a] | **3.28** | **0.02** | 0.96 |
| **CD8 lymphocytes** | **3.30** | **0.00** | 0.77 |
| CD4/CD8 ratio | 2.93 | 0.03 | 0.83 |
| **CD20 lymphocytes** | **3.12** | **0.02** | 0.76 |
| Monocytic IL-1 [b] production | 1.20 | 0.46 | 0.65 |
| Monocytic IFN-γ [c] production | 1.20 | 0.50 | 1.07 |
| Humoral immune parameters (Ig[d]G titers) | | | |
| *A.a.* (SUNY67) [e] | 0.52 | 1.00 | 1.30 |
| *A.a.* (Y4) | 2.38 | 0.38 | 1.63 |
| ***A.a.* (ATCC29523)** | **3.08** | **0.04** | 0.15 |
| *F.n* (ATCC 25586) [f] | 0.98 | 0.67 | 1.22 |
| *T.d.* (ATCC 35405) [g] | 1.77 | 0.60 | 1.58 |
| *P.i.* (ATCC 25611) [h] | 0.60 | 0.70 | 1.64 |
| *P.n.* (ATCC 33563)[i] | 2.29 | 0.52 | 1.59 |
| *C.o.* (S3) [j] | 2.99 | 0.60 | 0.80 |
| *E.c.* (ATCC 23834) [k] | 1.64 | 0.24 | 1.76 |

[a] CD = cluster of differentiation

[b] IL = interleukin

[c] IFN-γ = interferon

[d] Ig = immunoglobulin

[e] A.a. = *Aggregatibacter actinomycetemcomitans*

[f] F.n. = *Fusobacterium nucleatum*

[g] T.d. = *Treponema denticola*

[h] P.i. = *Prevotella intermedia*

[i] P.n. = *Prevotella nigrescens*

[j] C.o. = *Capnocytophaga ochracea*

[k] E.c. = *Eikenella corrodens*

occurring exacerbations in EOP patients than in LOP patients. Our strategy in anomaly detection was based on large sample entropy values and low clustering importance scores detected by unsupervised clustering of the patients. The two methods have no elements in common, but were found to be in concordance in detecting hidden complexity in the datasets.

Anomalies are difficult to detect in a dataset. Systems evolve over time and what qualifies as an anomaly first might change later. Anomalies of a given size will tend to be harder to detect in parameters with large variance, as compared to parameters with small variance [22]. The boundaries between normal and abnormal behavior are often not precise. The advantage of the current study is the relative "clear" labeling of the patients, which in general requires substantial effort to obtain. Sample entropy calculations in the validation cohort dataset were suggestive for anomalies in the neutrophil chemotaxis parameter. Other anomalies either never existed or if existed, they were no longer identifiable. The validation cohort is certainly a group

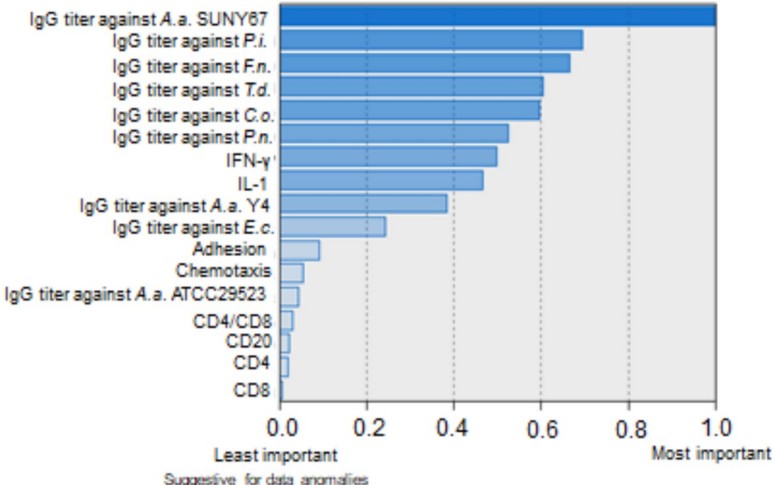

**Fig 3. Clustering importance evaluation of principal component analysis (PCA)-residual subspace parameters.** Ranking of the 17 PCA-residual subspace parameters according to their overall clustering importance in separating patients into two classes by the two-step clustering method in an unsupervised way. Low clustering importance of a parameter is suggestive for data anomalies.

of patients with severe disease (stage 3), but with a mean age higher than the EOP group. We can assume that anomalies can be found for a period of time and over the years the situation might change, perhaps due to treatment interventions. The smaller number of patients in the validation cohort might have prevented anomalies to be revealed.

On a population level, bistability is observed by two modes (peaks) in probability density distributions. We found in a previous study [6] on unlabeled periodontitis patients well-maintained over 5 to 8 years, possible evidence of periodontitis being a bistable system (showing

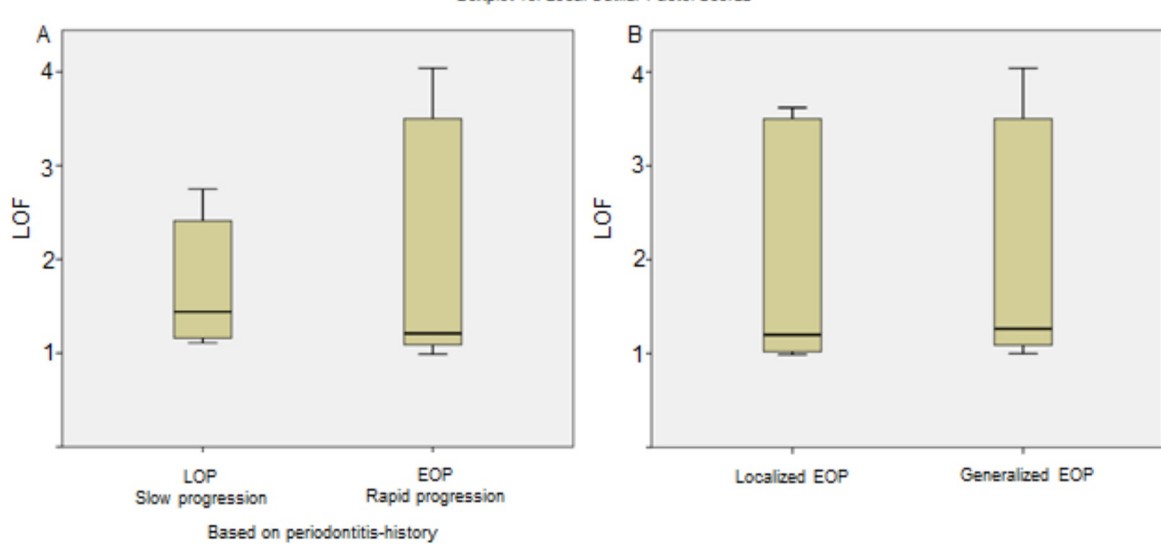

**Fig 4. Boxplot for Local Outlier Factor (LOF) scores among early-onset (EOP) and late-onset periodontitis (LOP) patients.** Anomalies in data present with higher LOF scores. Minimum, first quartile, median, third quartile and maximum values are shown for A. All EOP and LOP patient categories, B. Localized and generalized EOP patient sub-categories.

two main stable states). The smaller cluster showed radiographic bone loss level change 5 times more at average than the bigger cluster. Random fluctuations in immunologic parameters might push a nonlinear system (like periodontitis) from one state to the other [16]. Thus our current findings support the concept that EOP patients with rapidly progressive periodontal breakdown, having their "basal" set of causality factors, might convert more often and more severely in an exacerbation phase before the system regresses in a resolution (remission) phase [2]. A recent study identified three clusters of periodontal patients (phenotypes) on the basis of clinical, radiographic and microbiological data [23]. Finding pathophysiological pathways and our understanding of the periodicity of the disease, might identify endotypes within phenotypes, which in turn might enhance our prognostic and therapeutic abilities in clinical practice.

The hypothesis that stochastic gene expression has a significant effect on the biology of organisms was based on the observation that genetically identical organisms, maintained in identical environments, diverge phenotypically [16,17]. Fundamentally, this is because the expression of a gene involves the discrete and inherently random biochemical reactions involved in the production of mRNAs and proteins [16]. Fluctuations do not average away, but rather lead to differences in the function of otherwise identical cells [17]. In an alternative hypothesis, the stochastic kinetics of gene activity may be genetically determined by the promoter variation, which dictates various regulatory elements like histones and transcription factors, how to bind and unbind to their corresponding binding sites [24]. In this respect, epigenetic modifications of the genome, can equally be contributing to altered promoter activity and cause genes to behave in an aberrant way [25]. It must be noted that the current study was conducted on patients with a distinct genetic/epigenetic background (Japanese) and therefore extrapolating the results further to other populations needs to be performed with caution.

Predictive models when properly trained and tested (validated) can be applied in detecting anomalies [22] and thus identify potential periodontal patients to develop EOP or patients in an early stage of EOP. This could be helpful in a clinical setting, where EOP patients are considered more difficult and demanding to treat. Subtle changes detected in an early phase might give a warning signal of what could follow and preventive and treatment protocols may be started. Future studies on a wider array of parameters might reveal anomalies from unexpected sources. However, supervised modes of detection are less flexible in catching new anomalies as they cannot automatically adapt to new patterns [22]. We showed in previous studies on the sample used in the current study, that a supervised classification by decision trees [4] and artificial neural networks [5] could discriminate EOP from LOP. However, a correlation of predictive parameters to periodontitis, does not imply causation [26] and it only reflects the clinical status of the patients without providing prognosis. The current study suggests that we can go one step further and predict an ongoing or upcoming exacerbation of periodontitis. However, our LOF approach in predicting EOP provided results that could be generalized with caution due to a relatively high false positive rate (17%). Nonetheless, the high false alarm rates are always a problem in detecting anomalies [22].

*P.g.* has been reported as a keystone pathogen in periodontitis [27] and IgG titer against *P.g.* is reported in the current study as the first of the principal components in explaining the variance of the aggregate EOP and LOP sample. Monocyte IL-2 and IL-4 production are also found among the principal components in PCA and are reported IL-2 as significantly higher and IL-4 as significantly lower in LOP patients compared to EOP by mean values (Table 2). The central roles of IL-2 in regulating lymphocytes and of IL-4 in suppressing inflammation have been well studied [28]. The fact that an aberrant immune response in periodontitis constituting a state of hypo- or hyper-response or the inability to resolve properly inflammation, is connecting with the current identified parameters in a nonlinear fashion, explains the complex picture we are receiving [2]. In another example, IFN-γ considered the main phagocyte-

activating cytokine, was found in the current study significantly higher in EOP patients, but also found to belong to the sub-space parameters contributing zero in explaining the variance in the sample. The same situation applies to IgG titer for *C.o*., which was significantly higher in EOP, but also was found to belong to the PCA-subspace parameters. No indications for anomalies were found for all tested PCA-subspace IgG titers except for IgG titer against *A.a*. (Fig 3). *A.a*. has been associated with EOP and especially with the localized form of the disease [29]. The presence of *A.a*. in the oral cavity of young individuals increases the risk for initiation and progression of the disease [30]. However, it is accepted that the microbial composition of the subgingival biofilm cannot discriminate EOP from other periodontitis cases [31]. Antibodies against suspected periodontal pathogens are thought to clear out bacteria and significantly elevated levels of serum antibodies against *A.a*. have been found in EOP cases [32]. A pre-clinical role of *A.a*. has also been described. As periodontitis advances, the subgingival ecosystem becomes more anaerobic and more diverse [2,33]. Thus, *A.a*. may become more prevalent in the subgingival ecosystem, and an anomalous IgG titer against *A.a*. leaves space for *A.a*. to exert its pathogenic potential to host immune cells (e.g. via leukotoxin activity) resulting in worsened inflammation and concomitant tissue destruction.

Neutrophils are in the first line of defense against the dental biofilm bacteria and they express a large variety of cell surface receptors to sense the inflammatory environment [34]. The importance of CD4 lymphocytes in the immune response has been extensively studied, while the role of CD8 lymphocytes is not fully understood [35]. We found suggestive evidence in the current study that fundamental immune protective mechanisms like neutrophil chemotaxis and lymphocyte counts of CD4, CD8 and CD20 might be subject to random fluctuations that might result in the rapid progression of EOP. One obvious limitation of the current study originates from the fact that it is cross-sectional and as of that it is unknown how parameters might change in time. The changes that might appear in the anomaly status as a result of treatment is unknown, and therefore a confounding factor in the study might be a history of previous treatment.

This study introduces to periodontitis pathogenesis the well-accepted phenomenon of noise induced phenotypic variation due to stochasticity. By better understanding the mechanisms underlying the clinical expression of periodontitis and by developing predictive models that intercept incoming disturbing anomalies, we might be able to enhance our ability to cope with EOP. When biologically relevant combinations of microbial/immunological/genetic biomarker packages will be available for use in the future, overlaying artificial intelligence algorithms might warn patients to visit the periodontist since an exacerbation with rapid progression of periodontal support is upcoming or ongoing. The personal prediction of risk for disease exacerbation by applying artificial intelligence is currently being explored in other chronic diseases [36,37].

# Materials and methods

## Ethics statement

The Okayama University Dental Hospital committee approved the study [21]. Periodontitis patients were recruited as they presented at the Okayama University Dental Hospital over a period of 10 years. Informed written consent for taking blood for laboratory examination was obtained from each subject.

## Study population

We derived data from 162 Japanese periodontitis patients [21] (48 male and 114 female systemically healthy with a mean age 34.6 ± 12.2 years). The raw data set of the 162 patients was

used before in studies to explore mathematical models for periodontitis [5,6]. The following parameters were available: neutrophil chemotaxis, phagocytosis and adhesion to nylon fibers, T-cell blastogenesis against anti-CD3 monoclonal antibodies and pokeweed mitogen, as well as counts of CD3, CD4, CD8, CD4/CD8 ratio and CD20 lymphocytes in peripheral blood. In addition we used data of IL-1, IL-2, IL-4, IL-6, TNF-α and IFN-γ levels produced by mononuclear cells from peripheral blood. We also retrieved data from the same patients for serum IgG titers (assessed by enzyme-linked immunosorbent assay (ELISA)) against *Aggregatibacter actinomycetemcomitans* (*A.a.*) (Y4 antigen), *A.a.* (ATCC29523), *A.a.* (SUNY67), *Porphyromonas gingivalis* (*P.g.*) (FDC381), *P.g.* (SU63), *Eikenella corrodens* (ATCC23834) (*E.c.*), *Prevotella intermedia* (ATCC25611) (*P.i.*), *Prevotella nigrescens* (ATCC33563) (*P.n.*), *Capnocytophaga ochracea* (S3) (*C.o.*), *Wolinella succinogens* (ATCC29543) (*W.s.*), *Treponema denticola* (ATCC35405) (*T.d.*) and *Fusobacterium nucleatum* (ATCC25586) (*F.n.*). We obtained 68 EOP (localized and generalized cases aggregated) (mean age 26.2 ± 7.0 years) (stage 3 or 4 with grade C) and 43 LOP (mean age 47.0 ± 11.0 years) (stage 3 with grade A) cases for the discovery analysis. Another group of 51 patients were declared "suspected for EOP"; they had periodontitis stage 3 with grade C (mean age 36.0 ± 9.2 years). These patients were used as a validation cohort.

## Laboratory procedures

Cytokine productivity by T-cells was measured after in vitro stimulation with anti-CD3 monoclonal antibody. The amounts of secreted cytokines in the culture supernatants were made using radioimmunoassay for IL-1, IL2 and IFN-γ and ELISA for IL-4, IL-6 and TNF-α. Two color flow cytometric analysis using panels of monoclonal antibodies was employed to determine lymphocyte subsets. T-cell blastogenesis was evaluated by the uptake amount of thymidine ($^3$H). Antibody responses to periodontal bacteria were assessed by the ELISA technique. The correlation coefficient for the line fitting was above 0.90. Neutrophils were isolated from heparinized peripheral venous blood by discontinuous density gradient centrifugation. Neutrophil chemotaxis was assessed using N-formyl-methionyl-leucyl-phenylalanine, neutrophil phagocytosis was estimated by the number of bacteria internalized by 100 neutrophils and neutrophil adhesion was determined using a tuberculin syringe nylon fiber column that allowed blood to flow through by gravity.

## Statistical analysis

We compared means of immunologic parameters between EOP and LOP patients using the Mann-Whitney U test with a level of statistical significance set at < 0.05.

## Subspace analysis

Each dataset has its typical variation. However, there might be unusual conditions deviating from the typical variation [38]. We searched for collective anomalies, which is the term used when data instances (i.e. collected parameter values) are anomalous with respect to the entire dataset. The cut-off level of the typical variation and therefore the subspace region, can be determined by principal component analysis (PCA) [39]. Therefore PCA was applied on the cellular and humoral (serum IgG titers) immunologic parameters. After extracting the principal parameters that explain the vast majority of the variance of the data (EOP and LOP aggregated) and thus designating the normal variation, i.e. overall susceptibility, the remainder of the parameters were considered part of the residual subspace into which anomalies can be detected [39].

Deviation from the normal was searched by computing the sample entropy for each parameter in the residual PCA-subspace (after normalizing the data), a metric that captures the

degree of dispersal or concentration of a distribution [40]. Sample entropy is a sensitive metric for detecting and classifying changes in parameter distributions with a very low false positive rate. When all observations are the same, sample entropy takes the value of 0. On the other hand, high sample entropy values indicate anomalies. To calculate sample entropy we used the formula [40],

$$H(x) = -\sum_{i=1}^{N}\left(\frac{n_i}{S}\right)\log\left(\frac{n_i}{S}\right)$$

where $x = \{n_i, i = 1. . . ..,N\}$ and $S$ the total number of observations. The maximum value it can take is log ($N$). Entropy tends to increase as sample sizes increase.

We tested the performance of this approach of anomaly detection, through grouping the patients into two classes by the two-step clustering method using the newly identified PCA-subspace parameters as predictors [41]. The two-step clustering method uses both partitional (k-means) for an initial separation of patients and subsequently hierarchical (agglomerative) algorithms. The idea is that parameters with anomalies will confer lower overall clustering importance scores in unsupervised grouping of patients based on log-likelihood distance [41].

Additionally we computed the sample entropy of the residual PCA-subspace parameters for the validation cohort. The purpose of using this cohort was to disclose trends in sample entropy on parameters identified in the discovery cohort belonging to the residual PCA-subspace.

We finally set out to test the performance of the local outlier factor approach (LOF) in parameters with anomalies, to correctly classify EOP and LOP patients. The LOF algorithm assigns an aggregate "outlier" score for each individual in the dataset based on local density calculations [42]. Values outlying relative to their local neighborhoods, particularly with respect to the densities of the neighborhoods, are regarded as "local" outliers. LOF scores are ratios of the density of the neighborhood over the density of local outliers. Anomalies in data result in larger than 1 LOF scores, because outliers show low local densities compared to their neighbors [42]. A k-nearest neighbor classifier (k-NN) was used to identify EOP and LOP patients on the basis of the aggregate LOF scores.

We used SPSS version 20.0 programme (IBM, Chicago, IL, USA) to carry out the above described analyses and WEKA software (version 3.8.1; The University of Waikato, Hamilton, New Zealand) for LOF and k-NN.

## Supporting information

**S1 Data Set File.**
(XLSX)

## Acknowledgments

The authors declare no conflict of interest.

## Author Contributions

**Conceptualization:** George Papantonopoulos, Marja L. Laine, Bruno G. Loos.

**Data curation:** Keiso Takahashi.

**Formal analysis:** George Papantonopoulos, Chryssa Delatola, Marja L. Laine.

**Investigation:** Chryssa Delatola.

**Methodology:** George Papantonopoulos, Bruno G. Loos.

**Supervision:** Keiso Takahashi.

**Writing – original draft:** George Papantonopoulos, Bruno G. Loos.

**Writing – review & editing:** Marja L. Laine.

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
