## [Decision Letter · Decision Letter 0]

23 Aug 2019

PONE-D-19-19387

Hidden noise in immunologic parameters might explain rapid progression in early-onset periodontitis.

PLOS ONE

Dear Dr Papantonopoulos,

Thank you for submitting your manuscript to PLOS ONE. After careful consideration, we feel that it has merit but does not fully meet PLOS ONE’s publication criteria as it currently stands. Therefore, we invite you to submit a revised version of the manuscript that addresses the points raised during the review process.

ACADEMIC EDITOR:  Your manuscript has been now evaluated by two experts in the field. While they both found the manuscript timely and novel, they have raised slight concerns about the validity of some of the statements and the analysis perspectives (needing more careful and rigorous approach in some specific areas indicated in the reviewers' comments, reviewer 2 and 1 respectively). Also, the referees required clarifications in the methodologies used for reproducibility. They also asked much more extensive discussion on the meaning of the results and their validation approaches. Again, both of the reviewers although thought the manuscript is very novel and likely is important for the field, they expressed concerns as to the some of the wording on the results being overly stated (they are specified in the review comments). Reviewer 2 also requested data availability since PLOS one mandates the authors to discard large data sets to a public domain. Please read the journal guidelines for authors for details.   

We would appreciate receiving your revised manuscript by Oct 07 2019 11:59PM. To enhance the reproducibility of your results, we recommend that if applicable you deposit your laboratory protocols in protocols.io, where a protocol can be assigned its own identifier (DOI) such that it can be cited independently in the future. For instructions see: http://journals.plos.org/plosone/s/submission-guidelines#loc-laboratory-protocols

We look forward to receiving your revised manuscript.

Kind regards,

Özlem Yilmaz, DDS, PhD

Academic Editor

PLOS ONE

Journal Requirements:

"Okayama University dental hospital committee".

i) Please amend your current ethics statement to confirm that your named institutional review board or ethics committee specifically approved this study.

ii) Once you have amended this/these statement(s) in the Methods section of the manuscript, please add the same text to the “Ethics Statement” field of the submission form (via “Edit Submission”).

3. Please amend the subsection category “[FOR JOURNAL STAFF USE ONLY]” for your manuscript. Unfortunately, this is not a valid category. At this time, please choose one or more subsections that best represent the topic(s) of your study.

Additional Editor Comments (if provided):

Reviewers' comments:

Reviewer's Responses to Questions

**Comments to the Author**

1. Is the manuscript technically sound, and do the data support the conclusions?

Reviewer #1: Yes

Reviewer #2: Yes

2. Has the statistical analysis been performed appropriately and rigorously? 

Reviewer #1: Yes

Reviewer #2: Yes

3. Have the authors made all data underlying the findings in their manuscript fully available?

Reviewer #1: Yes

Reviewer #2: No

4. Is the manuscript presented in an intelligible fashion and written in standard English?

Reviewer #1: Yes

Reviewer #2: Yes

5. Review Comments to the Author

Reviewer #1: This is a very timely and technically sound contribution to the complexity of periodontal disease pathogenesis. The principal author and other authors in the same team have previously published several papers on this topic. They have already introduced the major concepts. In this work, artificial intelligence concepts are being incorporated to a previously published and extensively studied data set. Takahashi (2001) paper that the data set has been used from already identified the distinctions. One should be careful of interpretations of this work however, as the study (although elaborate and extensive) is cross-sectional. Therefore, it is not known if any of these parameters would change in response to treatment and validated their relevance as biomarkers of disease.

The authors touch on the impact of this work briefly in the discussion by stating that "Any large series of parameters available in data sets with overlaying artificial intelligence algorithms might warn patients to visit the periodontist since an exacerbation with rapid progression of periodontal support is upcoming or ongoing." However, as they may also appreciate, this is quite an impossible task until biologically relevant combinations of microbial/immunological/genetic biomarker packages are readily available for home use. Until then, the work and similar studies would represent as scientific contributions; and in this case, valuable applications of novel mathematical models to periodontal disease pathogenesis. Therefore, one should be very careful and perhaps self-critical of the interpretation and impact of the data analyses.

Reviewer #2: The manuscript reports on the findings of an analytical approach in attempt to discriminate between early-onset and late-onset periodontitis (EOP and LOP), specifically, the existence of hidden random fluctuations, that is historically known as exacerbations or “flare ups” in the periodontal disease progression. The authors aimed to come up with an analytical modelling to decipher the “hidden” disorders in the datasets that normally cannot make an obvious distinction between two clinical phenomena. Overall, this is a very timely, and interesting analytical approach and report on the ongoing discussions on the molecular and genetic differences between aggressive periodontitis and chronic periodontitis. The study is well-designed and well-written. The following are several minor points to be considered before accepting for publication:

1. Methods/results: Although patient demographics have been published elsewhere, it should still be provided to give a overall picture for the readers.

2. While the research question is to discriminate between rapid and slow progressing periodontal disease (based on history), pooling localized and generalized cases might also implement variability and noise, as the authors define. Although the subject numbers might be small in localized and generalized groups, it is important to look at the data with that perspective as well.

3. “Entropy values on data for neutrophil chemotaxis, CD4, CD8, CD20 counts and serum IgG titer against A.a. indicated the existence of possible anomalies”. This is unclear what exactly it is referring to. Does this mean that these analyses are not suitable to discriminate between EOP and LOP, or does it mean that they explain the possible exacerbations seen during the course of the diseases? Please clarify.

4. The authors indicate that LOF presented relatively high sensitivity (94%) and specificity 83% in identifying EOP, while they conclude in Discussion that the LOF approach in predicting EOP should be generalized with caution due to a relatively high false positive rate (17%). This should be indicated in the Abstract.

5. Since no details are given regarding the analytical methods of immunological markers and cytokines, it is difficult to compare or comment on analytical errors or differences between groups. Perhaps, the authors can give a brief information about the analytical method used and the variation coefficient observed in those analysis.

6. It is obvious that this data represents a specific genetic and epigenetic population, the authors should also discuss the applicability of this approach on other populations and data or on individual cases in the clinic.

7. There is no extensive discussion on the results with validation group. Was the approach helped to clarify the diagnosis of this group as initially defined as “suspected” EOP?

6. PLOS authors have the option to publish the peer review history of their article (what does this mean?). If published, this will include your full peer review and any attached files.

Reviewer #1: No

Reviewer #2: No

---

## [Author Response · Author response to Decision Letter 0]

5 Oct 2019

Dear Prof. Yilmaz,

We must thank you and the two reviewers for the in depth review process and the comments they provided. We appreciate the reviewer’s concerns, they were very to the point and helpful, and we amended the manuscript as follows:

Academic editor

We now provide clarifications in the methodologies used for reproducibility and we discuss further on the meaning of the results and their validation approaches. We also changed the wording on the results that you found overly stated. 

Additional requirements

1. We ensure that your manuscript meets PLOS ONE's style requirements.

2. We amended the ethics statement as you indicated.

3. We amended the subsection category “[FOR JOURNAL STAFF USE ONLY]” for the manuscript.

4. We now provide a data set file as a supportive information file.

Reviewer 1

“One should be careful of interpretations of this work however, as the study (although elaborate and extensive) is cross-sectional. Therefore, it is not known if any of these parameters would change in response to treatment and validated their relevance as biomarkers of disease.”

Response: We agree with the reviewer and we now write in the Discussion (amended manuscript page 14), “One obvious limitation of the current study originates from the fact that it is cross-sectional and as of that it is unknown how parameters might change in time. The changes that might appear in the anomaly status as a result of treatment is unknown, and therefore a confounding factor in the study might be a history of previous treatment.”

“However, as they may also appreciate, this is quite an impossible task until biologically relevant combinations of microbial/immunological/genetic biomarker packages are readily available for home use.”

Response: We responded to reviewer’s concern by writing (amended manuscript page 15) “When biologically relevant combinations of microbial/immunological/genetic biomarker packages will be available for use in the future, overlaying artificial intelligence algorithms might warn patients to visit the periodontist…”.

Reviewer 2

1. Methods/results: Although patient demographics have been published elsewhere, it should still be provided to give an overall picture for the readers.

 Response: We now provide a table with patient demographics (Table 1).

2. While the research question is to discriminate between rapid and slow progressing periodontal disease (based on history), pooling localized and generalized cases might also implement variability and noise, as the authors define. Although the subject numbers might be small in localized and generalized groups, it is important to look at the data with that perspective as well.

 Response: We now provide in the new figure 4 separate distributions for localized and generalized EOP patients. We report in Results (amended manuscript page 9) that “By separating localized from generalized EOP patients we found LOF score distributions to be similar in the two categories, with the generalized EOP category showing a higher maximum value (Fig 4B).”

3. “Entropy values on data for neutrophil chemotaxis, CD4, CD8, CD20 counts and serum IgG titer against A.a. indicated the existence of possible anomalies”. This is unclear what exactly it is referring to. Does this mean that these analyses are not suitable to discriminate between EOP and LOP, or does it mean that they explain the possible exacerbations seen during the course of the diseases? Please clarify. 

 Response: We clarify now that the findings explain the possible exacerbations seen during the course of the disease. We write (amended manuscript page 9) “Entropy values indicated possible data anomalies for neutrophil chemotaxis, CD4, CD8 and CD20 counts and IgG titer against A.a. (ATCC29523) that might explain more regularly occurring disease exacerbations in EOP patients.” 

4. The authors indicate that LOF presented relatively high sensitivity (94%) and specificity 83% in identifying EOP, while they conclude in Discussion that the LOF approach in predicting EOP should be generalized with caution due to a relatively high false positive rate (17%). This should be indicated in the Abstract.

Response: we now indicate that in the abstract.

5. Since no details are given regarding the analytical methods of immunological markers and cytokines, it is difficult to compare or comment on analytical errors or differences between groups. Perhaps, the authors can give a brief information about the analytical method used and the variation coefficient observed in those analysis.

 Response: We wrote a new subsection in Materials and Methods which we title “laboratory procedures”.

6. It is obvious that this data represents a specific genetic and epigenetic population, the authors should also discuss the applicability of this approach on other populations and data or on individual cases in the clinic.

 Response: Thank you for your comment. We now write in the Discussion (amended manuscript page 12) “It must be noted that the current study was conducted on patients with a distinct genetic/epigenetic background (Japanese) and therefore extrapolating the results further to other populations needs to be performed with caution.” 

7. There is no extensive discussion on the results with validation group. Was the approach helped to clarify the diagnosis of this group as initially defined as “suspected” EOP?

Response: We write in the Discussion (amended manuscript page 11-12), “The validation cohort is certainly a group of patients with severe disease (stage 3), but with a mean age higher than the EOP group. We can assume that anomalies can be found for a period of time and over the years the situation might change, perhaps due to treatment interventions. The smaller number of patients in the validation cohort might have prevented anomalies to be revealed.”

Yours sincerely,

George Papantonopoulos

Corresponding author

---

## [Editor Report · Decision Letter 1]

18 Oct 2019

Hidden noise in immunologic parameters might explain rapid progression in early-onset periodontitis.

PONE-D-19-19387R1

Dear Dr. Papantonopoulos,

We are pleased to inform you that your manuscript has been judged scientifically suitable for publication and will be formally accepted for publication once it complies with all outstanding technical requirements.

With kind regards,

Özlem Yilmaz, DDS, PhD

Academic Editor

PLOS ONE
---

## [Editor Report · Acceptance letter]

23 Oct 2019

PONE-D-19-19387R1 

Hidden noise in immunologic parameters might explain rapid progression in early-onset periodontitis. 

Dear Dr. Papantonopoulos:

I am pleased to inform you that your manuscript has been deemed suitable for publication in PLOS ONE. Congratulations! Your manuscript is now with our production department. 

With kind regards,

on behalf of

Dr. Özlem Yilmaz 

Academic Editor

PLOS ONE